# Chlorhexidine Promotes *Psl* Expression in *Pseudomonas aeruginosa* That Enhances Cell Aggregation with Preserved Pathogenicity Demonstrates an Adaptation against Antiseptic

**DOI:** 10.3390/ijms23158308

**Published:** 2022-07-27

**Authors:** Uthaibhorn Singkham-In, Pornpimol Phuengmaung, Jiradej Makjaroen, Wilasinee Saisorn, Thansita Bhunyakarnjanarat, Tanittha Chatsuwan, Chintana Chirathaworn, Wiwat Chancharoenthana, Asada Leelahavanichkul

**Affiliations:** 1Department of Microbiology, Faculty of Medicine, Chulalongkorn University, 1873 Rama 4 Road, Pathumwan, Bangkok 10330, Thailand; singkhamin.u@gmail.com (U.S.-I.); pphuengmaung@gmail.com (P.P.); chinchula99@gmail.com (C.C.); 2Center of Excellence in Systems Biology, Faculty of Medicine, Chulalongkorn University, 1873 Rama 4 Road, Pathumwan, Bangkok 10330, Thailand; jiradejmak@gmail.com; 3Translational Research in Inflammation and Immunology Research Unit (TRIRU), Department of Microbiology, Faculty of Medicine, Chulalongkorn University, 1873 Rama 4 Road, Pathumwan, Bangkok 10330, Thailand; wsaisorn@gmail.com (W.S.); thansitadew@gmail.com (T.B.); 4Antimicrobial Resistance and Stewardship Research Unit, Department of Microbiology, Faculty of Medicine, Chulalongkorn University, 1873 Rama 4 Road, Pathumwan, Bangkok 10330, Thailand; tanittha.c@chula.ac.th; 5Tropical Nephrology Research Unit, Department of Clinical Tropical Medicine, Faculty of Tropical Medicine, Mahidol University, Bangkok 10400, Thailand

**Keywords:** *Pseudomonas aeruginosa*, chlorhexidine, cross-resistance, biofilms, *Psl*, cell aggregate, wound

## Abstract

Because *Pseudomonas aeruginosa* is frequently in contact with Chlorhexidine (a regular antiseptic), bacterial adaptations are possible. In comparison with the parent strain, the Chlorhexidine-adapted strain formed smaller colonies with metabolic downregulation (proteomic analysis) with the cross-resistance against colistin (an antibiotic for several antibiotic-resistant bacteria), partly through the modification of L-Ara4N in the lipopolysaccharide at the outer membrane. Chlorhexidine-adapted strain formed dense liquid–solid interface biofilms with enhanced cell aggregation partly due to the Chlorhexidine-induced overexpression of *psl* (exopolysaccharide-encoded gene) through the LadS/GacSA pathway (c-di-GMP-independence) in 12 h biofilms and maintained the aggregation with SiaD-mediated c-di-GMP dependence in 24 h biofilms as evaluated by polymerase chain reaction (PCR). The addition of Ca^2+^ in the Chlorhexidine-adapted strain facilitated several *Psl*-associated genes, indicating an impact of Ca^2+^ in *Psl* production. The activation by Chlorhexidine-treated sessile bacteria demonstrated a lower expression of *IL-6* and *IL-8* on fibroblasts and macrophages than the activation by the parent strain, indicating the less inflammatory reactions from Chlorhexidine-exposed bacteria. However, the 14-day severity of the wounds in mouse caused by Chlorhexidine-treated bacteria versus the parent strain was similar, as indicated by wound diameters and bacterial burdens. In conclusion, Chlorhexidine induced *psl* over-expression and colistin cross-resistance that might be clinically important.

## 1. Introduction

Hospital-acquired bacterial infections caused by nosocomial pathogens are a concerning crisis due to high mortality, increased healthcare costs, and emerging antibiotic resistance [1]. As such, *Pseudomonas aeruginosa* has recently emerged as an important pathogen, causing chronic infections, partly due to the biofilm formation that is presented in several diseases; for example, cystic fibrosis, catheter-induced infection, and chronic wounds [1,2]. Because biofilms are bacterial products used for the protection against harsh environments (such as starvation, drought, and antibiotics [3]), biofilm production is one of the important mechanisms causing antibiotic resistance. In the initial step of biofilm formation, *P. aeruginosa* attaches to the surface with appendages (such as pili and outer membrane proteins), triggering microcolony forming on the surface. Then, the attached *P. aeruginosa* secrete exopolysaccharides (EPS) with 3 molecular components, including Pel (pellicle), alginate, and Psl (polysaccharide synthesis locus) [4]. Both Psl and Pel are important for the EPS initiation process follows by the production of alginate-rich polymer to develop the biofilm maturation [5,6]. While Pel and Psl are produced in the biofilms of both mucoid and non-mucoid *P. aeruginosa*, Pel is frequently presented in the air-liquid interface biofilms and alginate is a more predominant component in the mucoid than the non-mucoid biofilms [7]. Because the alginate-predominated biofilms have a looser matrix, there is an easier diffusion of the biofilm substances into the environment than the biofilms with Pel or Psl-dominance. In parallel, Psl-riched biofilms have a more densely packed matrix, protecting bacteria from harmful agents, than the alginate-based biofilms [7]. As such, the switching from alginate-predominated biofilms to Psl-based biofilms is associated with the alteration from acute to chronic *Pseudomonas* infections [8]. Accordingly, Psl not only increases biofilm rigidity but also induces bacterial cell aggregation [9] and promotes the chronic stage of infection, while alginate-dominant biofilms might be formed in the earlier phase of infection [8]. Hence, the external stimuli exposed by *P. aeruginosa* may play a crucial role in the variation of biofilm compositions and some stresses might push *Pseudomonas* biofilms toward the structures that are adapted for chronic infection through the denser biofilm matrix.

The most common stresses that *P. aeruginosa* frequently encounters in a healthcare setting are antibiotics and antiseptics [10]. Chlorhexidine digluconate (CHG) is a cationic broad-spectrum antiseptic that disturbs bacterial membranes [11]. As such, the bacterial membrane has a negative charge stabilized by divalent cations (Mg^2+^ and Ca^2+^) [12] which can be interfered with by Chlorhexidine through (i) the directs binding to the negative charge of lipopolysaccharide (LPS) and (ii) the displacement of divalent cations (LPS-stabilizing factor) with the precipitation of intracellular bacterial components [13]. The use of Chlorhexidine correlates with a reduced Chlorhexidine susceptibility and a cross-resistance to antibiotics, especially colistin, in *P. aeruginosa*, *Acinetobacter baumannii*, and *Klebsiella pneumoniae* [14,15]. Chlorhexidine, in sub-lethal concentrations, after the routine use of antiseptics leads to the adaptation of bacteria against Chlorhexidine and related antibiotics [16]. Notably, Chlorhexidine and colistin are positive charge molecules that bind to the negative charge of the bacterial membrane, which might explain the cross-resistance between Chlorhexidine and colistin [15,17]. Not only antibiotic cross-resistance but also a virulence-related adaptation in response to Chlorhexidine should be concerned. Indeed, several virulence genes (such as pili and efflux pumps) are also upregulated after the Chlorhexidine exposure leading to the cross-resistance against several antibiotics [15,18]. Because the sublethal concentration of Chlorhexidine is inevitable after each use in the healthcare environment, the use of Chlorhexidine might promote bacterial adaptation. Therefore, we conducted the prolonged exposure to sub-lethal Chlorhexidine situation in *P. aeruginosa* to investigate the phenotypic changes and the responses (proteomic analysis) with several testes, including in a wound infection mouse model. 

## 2. Results

### 2.1. Chlorhexidine Promoted Antibiotic Resistance and Colony Morphological Changes in P. aeruginosa That Related to the Alteration of Metabolic Activities 

Chlorhexidine-treated *P. aeruginosa* (C_PACL) is generated by the serial treatment of the clinical-isolated *P. aeruginosa* parent strain (PACL) (Figure 1A). In comparison with PACL, C_PACL formed smaller colonies, referred to as the small-colony variants (SCVs) (Figure 1B,C), and increased the resistance against Chlorhexidine, colistin, and imipenem, but not meropenem and tobramycin (Figure 1D). The minimal inhibitory concentrations (MICs) toward Chlorhexidine of C_PACL was two-fold higher than the parent strain (PACL) (from 4.88 to 9.77 mg/L). While PACL was susceptible to colistin (MIC = 0.5 mg/L), Chlorhexidine-exposed bacteria (C_PACL) was resistant to colistin (MIC = 4 mg/L). Likewise, the MIC to imipenem increased from 0.5 mg/L in PACL to 1 mg/L in C_PACL but the MICs to meropenem and tobramycin were not changed (Figure 1D). Because of the correlation among (i) the altered metabolic activities, (ii) the SCVs and biofilm formation, and (iii) the persistence and chronic infection in *P. aeruginosa* [19], the proteomic analysis between PACL versus C_PACL biofilms was performed. 

According to the proteomic study, 957 proteins were differentially expressed in C_PACL compared to PACL biofilms. Among these proteins, 496 proteins were upregulated, including 111 proteins with *p* < 0.05, 53 proteins with *p* < 0.01, and 22 proteins with *p* < 0.001 (Appendix A), and 461 proteins were downregulated, including 116 proteins with *p* < 0.05, 69 proteins with *p* < 0.01, and 29 proteins with *p* < 0.001 (Appendix A). The distribution of all upregulated and downregulated proteins is shown in the volcano plot (Figure 2A). The classifications of downregulated and upregulated proteins (*p* < 0.01) by the functional biological process are shown in Figure 2B,C. Most of the downregulated proteins were related to cellular metabolisms such as glycolysis, pyruvate metabolism, fatty acid synthesis, amide biosynthesis, and purine ribonucleotide metabolic processes, indicating the reduced metabolic rate in C_PACL (Figure 2B). The stress-induced energy starvation of C_PACL, possibly related to the smaller colonies of SCVs, was indicated by the down-regulation of several proteins for glycolysis, pyruvate metabolic process, and the TCA (tricarboxylic acid) cycle, including enolase (Eno), glyceraldehyde-3-phosphate dehydrogenase. (Gap), pyruvate dehydrogenase E1 component (AceE), dihydrolipoyllysine-residue acetyltransferase component of pyruvate dehydrogenase complex (AceF), fumarate hydratase (FumC2), phosphoenolpyruvate carboxylase (Ppc), dihydrolipoyllysine-residue succinyltransferase component of 2-oxoglutarate dehydrogenase complex (SucB), and phosphoenolpyruvate synthase (PpsA), together with enzymes in the fatty acid synthetic process, including fatty acid biosynthetic G, H, and Z (FabG, FabH, and FabZ) (Figure 2B and Appendix A).

Meanwhile, the upregulated proteins are associated with (i) antibiotic and Chlorhexidine resistance, including L-Ara4N (4-amino-4-_L_-deoxyarabinose), lipid A, and LPS biosynthetic processes and (ii) metabolic differences; such as cytochrome complex assembly and cellular polysaccharide biosynthesis (Figure 2C,D). Because L-Ara4N obscured the lipid A from colistin binding [20], the up-regulation of L-Ara4N modification (Arn) (ArnA, ArnB, ArnC, ArnD, and ArnT) and PmrA and PmrB (Arn expression regulators) [21] might be associated with the resistance. Additionally, the up-regulated TolA and TolB (Tol-Pal system proteins) in C_PACL might strengthen the outer membrane integrity through the increased lipid A biosynthesis (during stress) [22]. The increased maturation proteins (CcmE, CcmF, and CcmH) of cytochrome C (a terminal electron receiver in the aerobic respiratory chain [23]) in C_PACL suggested an adaptation toward the reduced cell energy status, while enhanced ubiquinone biosynthesis (UbiA and UbiG), an essential electron carrier in denitrification in *P. aeruginosa* [24], implied the responses against reactive radical species in C_PACL. Moreover, Chlorhexidine might also reduce bacterial virulence in C_PACL because there was a downregulation of the controllers of many virulence factors, including (i) *Pseudomonas* quinolone signal (PQS) synthesis (PqsA, PqsB, and PqsD), (ii) host invasion promotor; LasA (protease) and LasB (elastase), and (iii) phenazine biosynthetic proteins (PhzF, PhzM, and PhzS) (Appendix A). The protein connection network study demonstrated that the upregulated proteins were related to RecA, a pathway that correlated with the significant stresses or “SOS” response (a global response to DNA damage-inducing the cell cycle arrest, DNA repair, and mutagenesis) [25] in C_PACL when compared with PACL group (Figure 2D).

### 2.2. Alteration of P. Aeruginosa Extracellular Polysaccharides Leading to Cell Aggregation Possibly through Psl-Predominance after the Exposure to Chlorhexidine 

Because SCVs play a critical role in the biofilms of *P. aeruginosa* during chronic infection [26], we characterized the biofilm structure of C_PACL (Figure 3A–F). Indeed, C_PACL formed denser crystal violet-stained biofilms at the solid–liquid interface at the lateral surface of the well in polystyrene plates, indicating the increased strength of the interface biofilm when compared with PACL strain (Figure 3A,C). Moreover, the liquid-air interface biofilms, called pellicles, in PACL produced rough cracked pellicles, while C_PACL produced large fine pellicles (intact sheets), indicating the higher cell aggregate in the C_PACL pellicles (Figure 3B). With the confocal microscopy, PACL formed enormous extracellular matrix (ECM) biofilms (stained with red fluorescence, Figure 3D,F), indicating robust biofilm mass but a similar level of bacterial cells (stained with green fluorescence, Figure 3E,F). Conversely, C_PACL showed significantly less fluorescence-stained biofilm mass (Figure 3D,F), but there were densely packed bacterial cells in the biomass (Figure 3E,F), indicating an aggregate biofilm of C_PACL. Hence, Chlorhexidine exposure in C_PACL established a vigorous solid–liquid interface, prominent cell aggregation, and well-formed pellicle biofilms but not robust ECM production.

Among three extracellular polysaccharides (EPSs) produced by *P. aeruginosa* (Figure 4A), alginate is a major component of slime biofilms, while *Psl* and *Pel* maintain the strength of solid–liquid and the air-liquid interface, respectively [7]. Both PACL and C_PACL isolates formed a similar wrinkling colony biofilm but at the edge of wrinkling of C_PACL colonies exhibited more *Psl* and *Pel* than PACL as indicated by the stained color of Congo red (a color stained for Pel or *Psl* but not alginate) [27,28], particularly in 14-day biofilms (Figure 4B). These data suggested a possible higher binding capacity of C_PACL than PACL. In addition, a high Congo red binding capacity of C_PACL was confirmed in Luria-Bertani (LB) broth (Figure 4C,D). There was an upregulation of *pslB* and *pelA* (the genes encoding for *Psl* and *Pel*, respectively) in the 24-h biofilm of C_PACL than PACL without the difference in *algD* (an alginate-encoding gene) (Figure 4E–G), possibly associated with *Psl* and *Pel*-mediated denser cells [9]. Because the different expression of *pslB* between C_PACL and PACL was more prominent than the alteration in *pelA*, the enhanced aggregation during C_PACL biofilm-forming might be mostly due to *psl* than *pel* expression. 

### 2.3. Chlorhexidine Induced Psl-Predominated Biofilms in P. aeruginosa via SiaD and LadS/GacSA Regulatory System

Several “Surface attachments defective diguanylate cyclases” (such as SadC and SiaD; the control of SiaABCD complex for *psl* operon regulation [29,30]) through bis-(3′-5′)-cyclic dimeric guanosine monophosphate (c-di-GMP) [29,31], while LadS/GacSA (the hybrid histidine kinase recognizing several signals, including Ca^2+^) induces small RNA (*rsmZ* and *rsmY*) [32,33] to sequester RsmA (negative *psl* regulator through *psl* mRNA degradation [34]) in c-di-GMP-independent *Psl* production (Figure 5A). Hence, several genes of these pathways were explored (Figure 5B–J). At 12 h biofilms, the c-di-GMP-independent *Psl* biosynthesis [35] in C_PACL was more prominent than PACL as indicated by upregulation of *gacS*, *gacA*, *ladS*, *rsmZ*, *rsmY*, and *rsmA* (Figure 5B–G). Likewise, there was less motility and upregulation of *pilA* and *oprF* (the encoding genes for bacterial appendages used for cell attachment) (Figure 6A–D) with lower expression of *sadC* and *siaD* (the c-di-GMP-dependent *Psl* production) in C_PACL than PACL (Figure 5H–I). But the activity of c-di-GMP-independent *Psl* biosynthesis by these parameters of C_PACL decreased in 24 and 48 h biofilms (Figure 5B–G). Then, the c-di-GMP-dependent *Psl* biosynthesis of C_PACL was more prominent than PACL at 24 h biofilms as indicated by upregulation of *siaA* and *siaD* (but not *sadC*) (Figure 5H–J), implying the SiaD-mediated c-di-GMP-dependent of *Psl*-production at 24 h (Figure 5A). Notably, the upregulation of *rsmA* (a negative regulator of *Psl* production) was demonstrated only in 12 h of C_PACL biofilms, but neither in 24 nor 48 h of biofilm production (Figure 5G). These results suggested that Chlorhexidine enhanced *P. aeruginosa Psl*-predominated aggregate initially through c-di-GMP-independent LadS/GacSA at 12 h and the *Psl* production was sustained via SiaD-mediated c-di-GMP dependent pathway at 24 h and 48 h.

Because (i) the activation of c-di-GMP-independent LadS/GacSA is Ca^2+^ dependence [33], (ii) Ca^2+^ is one of the divalent ions stabilizers for the negative charge of LPS in Gram-negative bacteria [36], and (iii) Chlorhexidine replaces the divalent ions [13] that might release Ca^2+^ from the bacterial membrane and activate the inner membrane sensor (LadS and GacSA system) (Figure 5A), the importance of Ca^2+^ in *Psl* production is tested (Figure 7A). Indeed, Chlorhexidine with CaCl_2_ in combination showed a greater expression of *gacS*, *ladS*, and *pslB* than either Chlorhexidine or CaCl_2_ alone at 12 h biofilm production (Figure 7B–D), indicating a Chlorhexidine-Ca^2+^ synergy on the GacSA activation that subsequently enhanced *psl* upregulation. Perhaps, the free Ca^2+^ from the bacterial cell membrane, after Chlorhexidine cationic replacement, might turn on the LadS/GacSA-mediated *psl* operon, at least in part, through SiaD-induced c-di-GMP pathway (Figure 5A).

### 2.4. The Responses of Fibroblasts and Macrophages against Pseudomonas Parent Strain (PACL) and Chlorhexidine-Activated Strain (C_PACL) 

Because the *psl*-mediated aggregation in *P. aeruginosa* plays a critical role in chronic wound infection [37,38] which consists of several cell types, including fibroblasts and macrophages, the responses of these cells against both strains of *P. aeruginosa* are interesting. The difference between the fibroblast responses against sessile bacteria (bacteria with biofilms) versus planktonic form (bacteria without biofilms) within PACL parent strain or C_PACL was not obviously different. Accordingly, there was the upregulated *TLR-4* but downregulated *IL-6* without the difference in other genes nor cytokine responses in sessile PACL compared with planktonic PACL (Figure 8A–M). Meanwhile, there was a higher *TLR-4* and *TNF-α* expression but lower *IL-6, IL-8, TGF-β*, and *GM-CSF* in sessile C_PACL when compared with the activation by planktonic C_PACL (Figure 8A–M). Thus, the difference of the activation by sessile versus planktonic PACL or sessile versus planktonic C_PACL toward fibroblasts were not obvious. In comparison with planktonic PACL, the planktonic C_PACL upregulated *IL-8, TGF-β, GM-CSF, TLR-2, TLR-5*, and *TLR-6* but downregulated *IL-6* without the alteration in other parameters in fibroblasts (Figure 8A–M) implied a mildly prominent pro-inflammation of planktonic C_PACL over planktonic PACL. In contrast, the sessile C_PACL down-regulated *IL-6, IL-8, TGF-β,* and *GM-CSF* when compared with sessile PACL (Figure 8A–M) suggested a less inflammatory impact of C_PACL biofilms on fibroblasts possibly due to the reduced biofilm biomass in sessile C_PACL compared with sessile PACL (Figure 3A–F). These data indicated that Chlorhexidine decreased the pro-inflammatory properties of the sessile *P. aeruginosa* (but not the planktonic form) toward fibroblasts. In THP-1-derived macrophages, the sessile form PACL induced more inflammatory responses than the planktonic PACL as indicated by the higher expression of *TLR-2, TLR-4*, and *iNOS* (the pro-inflammatory molecules) with lower expression of *Arg-1* and *IL-10* (the anti-inflammatory molecules) without the alteration in other parameters (Figure 9A–O). The more prominent proinflammation of sessile PACL over planktonic PACL on macrophages supports a pro-inflammatory impact of biofilm structures on macrophages [39]. Meanwhile, the sessile C_PACL induced less inflammation than the planktonic C_PACL as indicated by the lower expression of *TNF-α, IL-8,* and *iNOS* (but lower *IL-10*) (Figure 9A–O) suggesting that the biofilm of C_PACL induced some anti-inflammatory effects on macrophages, possibly *Psl* or *Pel*-induced anti-inflammation [39,40]. Additionally, the planktonic C_PACL induced lower macrophage inflammatory responses than planktonic PACL as indicated by lower expression of *TLR-2, TLR-4, TLR-5, TLR-6, IL-6,* and *iNOS* with higher *IL-10* expression (Figure 9A–K) supporting the less prominence pro-inflammation of C_PACL bacteria without the biofilm molecules. Likewise, the sessile C_PACL induced lower macrophage inflammatory responses than sessile PACL as indicated by lower expression of *TLR-6, TNF-a, IL-6, IL-8,* and *iNOS*; however, with the lower *Arg-1* and *IL-10* expression (Figure 9A–K). These data indicated that Chlorhexidine decreased the pro-inflammatory properties of both sessile and planktonic *P. aeruginosa* toward macrophages.

### 2.5. Psl-Predominated Chlorhexidine-Activated Strain (C_PACL) Similarly Induced Wound Infection in Mice When Compared with the Parent Strain (PACL)

From the in vitro experiments, Chlorhexidine reduced *Pseudomonas* biofilm abundance (Figure 3) with increased *Psl*-predominance (Figure 4), partly through Ca^2+^-dependent operon activation (Figure 5 and Figure 7) through several signals (Figure 5 and Figure 7), resulting in a decrease in the responses of fibroblasts and macrophages (Figure 8 and Figure 9). Then, we tested both strains on a wound model to explore bacterial pathogenicity (Figure 10A). As such, the severity of infection as determined by wound size and bacterial burdens using C_PACL was lower than PACL at 7 days of experiments, but not at 3 and 14 days, without the difference in serum cytokine levels at 14 days of experiments (Figure 10B–H). The infection by a single type of organism was supported by the similar colony morphology (flat, opaque, and green-pigment colonies with irregular margins) with Gram-negative bacilli from the culture of wound fluid (Appendix A) and the bacterial colonies from the wound fluid of C_PACL infection were smaller than the PACL colonies, supporting Chlorhexidine-induced small variant colonies. Despite no bacteremia in all groups (data not shown), serum IL-1β, IL-6, and TNF-α (but not IL-10) were increased in the infected mice from both bacterial strains (Figure 10E–H). On the 14th-day experiment, the wound severity score (abundance of infiltrated immune cells beneath the wound beds, depth, and wellness) of C_PACL infection was similar to the PACL group using H&E staining (Appendix A). Our data indicated that Chlorhexidine decreased *Pseudomonas* virulence as indicated by biofilm biomass, host cell responses (fibroblasts and macrophages), and the virulence in mice (at 7 days of the wounds), despite an elevated Psl production. Although Chlorhexidine did not enhance bacterial virulence, the induction of antibiotic resistance should be concerned. With the exposure of Chlorhexidine on the other 12 clinical-isolated strains of *P. aeruginosa*, the increased MICs against Chlorhexidine or colistin was detected in 8 and 10 strains, respectively, MICs against both agents increased in 6 strains, and small colonies (SCVs) was demonstrated in 6 strains (Figure 11A–C and Appendix A). Hence, our study strongly indicated that the use of Chlorhexidine could induce *P. aeruginosa* with pathogenicity and antibiotic cross-resistance (especially against colistin) which might be clinically important.

## 3. Discussion

### 3.1. Chlorhexidine Use and Colistin Cross-Resistance

Although Chlorhexidine, a frequently used antiseptic in hospital, effectively control bacteria by the chemical properties [41] and antiseptic reservoir (sponge effect of the killed bacteria that destroys other bacterial generations referred to as the zombie effect of Chlorhexidine remnants) [42], the sub-lethal Chlorhexidine concentrations after each use will naturally be lower than the bactericidal levels and possibly induce bacterial adaptations [41]. Indeed, Chlorhexidine remnant replaces Ca^2+^ and penetrates the phospholipid layer of the *P. aeruginosa* membrane [13] with several evasion mechanisms to increase bacterial membrane integrity against the antiseptic through various proteins (OprF, TolA, and TolB) [15] supported the *TolA* and *TolB* upregulation in our Chlorhexidine-stimulated bacteria. Due to similar properties and actions of Chlorhexidine and colistin, colistin cross-resistance is frequently demonstrated following Chlorhexidine exposure [15,18], possibly through L-Ara4N modification in lipid A (a binding target of colistin [17]) of the bacterial membrane [43]. In colistin-resistant *P. aeruginosa*, the colistin-modified lipid A can be reformed by adding the positively charged L-Ara4N [43] or enhanced lipid A biosynthesis or increased efflux pump functions [15,18,44]. Here, increased L-Ara4N was demonstrated in proteomic analysis of Chlorhexidine-treated *P. aeruginosa.*

### 3.2. Chlorhexidine Use Induced Small Colony Variants (SCVs) with Psl-Predominated Biofilms through c-di-GMP-Independence and -Dependence at 12 and 24 h of Biofilms, Respectively

The *P. aeruginosa* SCVs (a small colony with 1–3 mm in diameter firstly isolated from patients with cystic fibrosis at 24–48 h of 37 °C incubation [45]) demonstrated metabolic arrest as indicated by the reduction of several proteins in proteomic analysis. Other characteristics of SCVs, including wrinkle, auto-aggregation, hyper-adherence, and biofilm formation as previously described [46], were also demonstrated in Chlorhexidine-treated strain (partly associated with *Psl*, a neutral pentasaccharide for biofilm initiation, attachment, and microcolony formation [4]). The cell aggregation through *Psl*-mediated polymer bridging manifests a less order alignment that intensely packed cells at the center of biofilms [9] similar to our Chlorhexidine-treated strain (Figure 3F). Indeed, the *psl* operon is composed of 15 co-transcribed genes (*pslA* to *pslO*) encoding essential proteins for Psl biosynthesis [7] that are regulated by c-di-GMP [29] through many *P. aeruginosa* diguanylate cyclases (SadC and SiaD) [30,47] or non-c-di-GMP pathways [34]. In c-di-GMP-independent *Psl* production, RsmA (a repressor protein that switches off *psl* [34]) is sequestered by *rsmY* and *rsmZ* [48], and controlled by GacSA (a positive regulator activating by the phosphorylation using GacS and LadS kinases [32] that is partly stimulated by Ca^2+^ [33,35], thiamine, and other metabolites [49]). Although the role of c-di-GMP on the *psl* promoter is still unclear due to the non-predicted *Psl*-binding site found in SadC and SiaD [30], the addition of exogenous *Psl* in a previous publication promotes *psl* expression implying *Psl* positive feedback on SadC and SiaD [30]. In our study, C_PACL biofilms overexpressed *pslB* (Figure 4F) related to *siaD* (Figure 5I) with an increased binding affinity of Congo red (the color stained for *Psl* or *Pel*) suggesting that *Psl* expression was partially maintained by c-di-GMP, especially in 24-h biofilms. On the other hand, LadS/GacSA (the c-di-GMP-independent *Psl* synthesis) was important at 12 h biofilms of C_PACL which could be facilitated by Ca^2+^ (a universal signal used for cellular responses [50]) possibly as the switching from an acute to a chronic adaptation [33]. Due to the Ca^2+^ displacement and the loss of membrane integrity by Chlorhexidine, Ca^2+^ might be released from the membrane and activates LadS/GacSA (Figure 5A) which is responsible for the initial adaptation against Chlorhexidine. More studies on this topic are interesting.

### 3.3. Chlorhexidine-Treated Pseudomonas Caused Chronic Wound in Mice, despite a Mildly Reduced Disease Severity, with a Possibility of Antibiotic Cross-Resistance

*P. aeruginosa* biofilm with alginate and *Psl* predominance has been demonstrated in cystic fibrosis (a chronic infection) [2,38] and *Psl* causes treatment failure partly through antibiotic tolerance from intense cell aggregation [9,38]. In contrast, there are only a few studies of *P. aeruginosa* biofilms in wound infections. In the infected wounds [51], fibroblasts mediate tissue homeostasis [52] and recognize pathogen-associated molecular patterns (PAMPs) by several TLRs with other mediators such as IL-6, IL-8, and MMP-1 (matrix metalloproteinase-1) [53]. Meanwhile, macrophages professionally recognize PAMPs and produce several cytokines (such as IL-6) that also promote wound healing through the proliferation of several cell types (fibroblasts, keratinocytes, and myofibroblasts) [54]. Although the expression of some inflammatory genes of fibroblasts and macrophages induced by the Chlorhexidine-activated strain was lower than the parent strain, the cytokine production after activation by both strains was not different. Because some molecules of *P. aeruginosa* biofilms (such as alginate) directly induce skin inflammation that might be a chemical interference of the wound model [55], the planktonic *P. aeruginosa* are applied in mouse wounds. Here, bacterial burdens in the wounds of C_PACL at 7 days were less than the PACL parent strain indicating a possible milder disease severity from Chlorhexidine-exposed bacteria, despite the non-difference at 3 and 14 days of the model. The Psl-dominant biofilms resist several immune responses (phagocytosis, neutrophil oxidative burst, and complement deposition) [56], and the Psl-overproduced *P. aeruginosa* PAO1 induces a higher bacterial load than the regular strain in 35 days porcine wounds [26]. In contrast, the 14 days mouse wounds from Psl-producing and Psl-depleted *Pseudomonas* PAO1 are not different [40]. Then, impacts of Psl on the wounds are still inconclusive. Despite the non-difference of Chlorhexidine-exposed *Pseudomonas* to the parent strain in our wound model, the Psl-predominated *Pseudomonas* after Chlorhexidine use might induce more severe infections in other models, as determined from several Psl-mediated bacterial evasion mechanisms [56]. Moreover, the cross-resistance against colistin and imipenem after Chlorhexidine use might lead to further clinical problems because of the importance of both agents against the current antibiotic-resistant bacteria [57]. Because chronic exposure to low-dose Chlorhexidine is necessary for the development of Psl-mediated bacterial adaptation, the adequate doses with frequent use of Chlorhexidine might be beneficial for the clinical translation. More studies on this topic are interesting.

## 4. Materials and Methods

### 4.1. Animals and Animal Model

The Institutional Animal Care and Use Committee of the Faculty of Medicine, Chulalongkorn University has approved the Animal care and use protocol (SST012/2564) following the US National Institutes of Health guidelines using specific pathogen free mouse facility with a 12:12 h light/dark cycle at 22 ± 2 °C and 50% relative humidity. As such, 8-week-old male C57BL/6 mice (Nomura Siam International, Pathumwan, Bangkok, Thailand) were used with free access to water and food at the Faculty of Medicine, Chulalongkorn University for a mouse model of wound infection as previously described [58]. Under the isoflurane anesthesia (Baxter, Puerto Rico, USA), the skin at the back of mice (left and right sides) was shaved, cleaned with 10% povidone-iodine, and induced an open wound with an aseptic procedure using a skin puncture (a punch biopsy with diameter 8 mm) (Oem factory, Guangdong, China) before the inoculation of 1 × 10^6^ colony-forming unit (CFU) of *P. aeruginosa* (parent and Chlorhexidine-exposed strains; details below) in normal saline solution (NSS) with the total volume of 50 µL per wound or NSS alone. Then, the open wounds were covered with 3M Tegaderm films (1622W) (3M Science, USA) and the films were daily changed until sacrifice. After the wound induction, a single subcutaneous injection (25 mg/kg) of tramadol (Sigma- Aldrich, St. Louis, MO, USA) was performed for postoperative analgesia. Fluid material on the wound was collected on the 3rd, 7th, and 14th days of experiments for *P. aeruginosa* culture (representing bacterial burden in the wounds). Because the wound fluid was not enough for the direct pipetting, NSS (50 µL) was put on the wound before stirring with the pipette tip, and collection of 20 µL wound fluid for the culture in LB (Luria-Bertani) agar plates (the non-selective media) before colony enumeration after 24 h incubation at 37 °C. The infection by the single organism was supported by the staining of Gram-negative bacilli in all colonies. On day 14, all mice were sacrificed with cardiac puncture under isoflurane anesthesia before aseptic sample collection. The serum cytokine levels were determined by ELISA kit (Invitrogen, Carlsbad, CA, USA). After sacrifice, wound tissues were carefully collected and preserved in 10% formalin. The tissues were processed, cut into sections, and placed on a glass slide, followed by hematoxylin and eosin staining. The histopathological analysis was evaluated based on inflammatory cells infiltration, wound diameter, wellness, and redness (inflammatory degree evaluation) with a score (0–3) according to a previous publication [59].

### 4.2. Chlorhexidine and Antibiotic Susceptibility Test 

The clinical strains *P. aeruginosa* (PACL, PA1, PA2, PA4, PA5, PA6, PA7, PA9, PA10, PA11, PA12, PA13, and PA14) were isolated from the patients in the King Chulalongkorn Memorial Hospital, Bangkok, Thailand as approved by the Institutional Review Board of the Faculty of Medicine, Chulalongkorn University (IRB 610/2564) and the Institutional Biosafety Committee (MDCU-IBC001/2022). The minimal inhibitory concentrations (MICs) against Chlorhexidine digluconate (Sigma-Aldrich, Darmstadt, Germany), colistin (Sigma-Aldrich), imipenem (Siam Pharmaceutical, Bangkok, Thailand), meropenem (Pfizer, NY, USA), and tobramycin (Novartis, Basel, Switzerland) were evaluated by the broth microdilution method using cation-adjusted Mueller–Hinton broth (BBL, MD, USA) as recommended by the Clinical and Laboratory Standards Institute (CLSI) guidelines [60]. Briefly, serial dilutions of these agents were prepared in 96-well culture plates, followed by inoculation of *P. aeruginosa* isolates (final concentration 1 × 10^5^ CFU/well) and incubation at 37 °C for 18 h. The MICs, the lowest concentration that can inhibit bacterial growth, and the susceptibility test were visually evaluated following the CLSI guidelines (Appendix A). 

### 4.3. Chlorhexidine Induction Protocol

For Chlorhexidine induction, all 13 *P. aeruginosa* isolates were grown in LB broth (Difco, Rockville, MD, USA) containing Chlorhexidine at the concentrations 0.25-fold (0.25×) of the MIC at 37 °C with shaking at 200 rpm for 48 h. After incubation, 1:100 of bacterial culture was grown in LB broth supplemented with a serially increasing concentration of Chlorhexidine (0.25×, 0.5×, 1×, and 2×) at 37 °C with shaking at 200 rpm for 48 h. After induction, *P. aeruginosa* was sub-cultured daily on the Chlorhexidine-free LB agar plate for ten days. 

### 4.4. Biofilm Formation Study

Biofilms forming of *P. aeruginosa* parent strain (PACL) and Chlorhexidine (CHG)-treated strain (C_PACL) in polystyrene 96-well culture plates, on glass coverslips, or in glass flasks were evaluated. The turbidity of *P. aeruginosa* that were cultured overnight in LB broth was adjusted to 0.5 McFarland standard. In 96-well plates, bacteria were cultured for 12, 24, and 48 h of static incubation at 37 °C. before biofilm identification with crystal violet color following a previous publication [61]. Briefly, the planktonic bacteria were removed, and biofilms were gently washed with distilled water twice before staining with 0.1% crystal violet for 15 min and washed twice with distilled water. Similarly, biofilms on glass coverslips were prepared in 6-well culture plates containing bacterial suspension with a glass coverslip at the bottom before removing the suspension, washed with phosphate buffer saline (PBS), fixed with 10% formaldehyde for 10 min, and stained for bacterial DNA with SYTO 9 green fluorescence (Invitrogen, Carlsbad, CA, USA) for 30 min. Then, the biofilm matrix was stained with Alexa Fluor 647 conjugated of Concanavalin A (AF647) (Invitrogen, Carlsbad, CA, USA) for 30 min. The biofilm images were photographed by the confocal laser scanning microscope, LSM 800 with Airyscan (Carl Zeiss Microscopy GmbH, Jena, Germany). Bacterial DNA (green) and biofilm matrix (red) fluorescent stains were visualized with a 63×/1.4 oil immersion lens. A total of 6 Z-stack images of the biofilms were acquired and analyzed using Zen microscopy software (Carl Zeiss Microscopy GmbH, Germany). The fluorescent intensities were also calculated using Zeiss microscopy software.

### 4.5. Congo Red Binding Assay

The non-alginate components of polysaccharide (*Psl* or *Pel*) in *P. aeruginosa* biofilms were stained by Congo red in LB agar. Briefly, *P. aeruginosa* colony biofilms of *P. aeruginosa* parent strain (PACL) and Chlorhexidine (CHG)-treated strain (C_PACL) were formed on LB agar plus 1% Congo red for 14 days at 25 °C and visually observed (red color on the biofilms is *Psl*- or *Pel*-stained polysaccharide). In parallel, the Congo red binding (an ability of bacteria to bind with the color) was quantified using *P. aeruginosa* growing in LB broth supplemented with 40, 80, and 160 µg/mL of Congo red for 24 h at 37 °C with shaking. The bacterial cells were pelleted and the remaining Congo red color in the culture supernatant was measured using OD_495nm_ (Congo red detection). The Congo red binding was calculated by the initial OD_495nm_ of the culture supernatant (the color intensity before capturing by bacteria) subtracting with OD_495nm_ of the culture supernatant after 24 h of bacterial growth (the color intensity after binding by bacteria).

### 4.6. Bacterial Gene Expressions

The expression of genes encoding for biofilm-related proteins of *P. aeruginosa* parent strain (PACL) and Chlorhexidine (CHG)-treated strain (C_PACL) was determined by quantitative real-time reverse transcription-polymerase chain reaction (qRT-PCR) using primers listed in Appendix A. Total RNA of planktonic (bacteria without biofilms) or sessile (bacteria with biofilms) forms were extracted using TRIzol^®^ Reagent (Invitrogen, Carlsbad, CA, USA), and cDNA was synthesized using RevertAid First Strand cDNA Synthesis Kit (Thermo Scientific, Vilnius, Lithuania). The number of biofilm-related transcripts was quantified by the QuantStudio6 (Applied Biosystems, Foster City, CA, USA) using the SYBR^®^ Green PCR Master Mix (Applied Biosystems). The relative number of transcripts was normalized with *16S rRNA* and calculated using the 2^−ΔΔct^ method.

### 4.7. Bacterial Motility Testing

The motile activity of *P. aeruginosa* parent strain (PACL) and Chlorhexidine (CHG)-treated strain (C_PACL) was performed on 0.5% LB agar. Briefly, overnight culture of *P. aeruginosa* in LB broth was spotted at the center of the 0.5% LB agar plate. After incubation at 37 °C for 18 h, the bacterial growth wider than the initial spot indicated the ability to motile on the agar surface. In contrast, the bacterial growth at the initial spot demonstrated the motility defect.

### 4.8. Proteomic Analysis

The 24-h biofilms of the parent *P. aeruginosa* (PACL) and Chlorhexidine-activated *P. aeruginosa* (C_PACL) were prepared in glass flasks and removed by cell scrapers. The biofilm pellets were collected by centrifugation at 4500 rpm at 4 °C for 30 min, mixed with 8 M urea, sonication, and determining protein concentrations (BCA assay). Then, the protein mixtures were digested with trypsin and the peptides from C_PACL and PACL were labeled with light (CH_2_O and NaBH_3_CN) and medium (CD_2_O and NaBH_3_CN) dimethyl labeling reagents, respectively, before quenching the reaction with ammonia solution and formic acid. The labeled peptides were separated into 9 fractions using the Pierce High pH Reversed-Phase Peptide Fractionation Kit. Eluates of each fraction was dried in a SpeedVac centrifuge before LC-MS/MS analysis. The fractionated samples were resuspended in 0.1% formic acid and then analyzed via an EASY-nLC1000 system (Thermo Fisher Scientific, Waltham, MA, USA) coupled to a Q-Exactive Orbitrap Plus mass spectrometer (Thermo Fisher Scientific) equipped with a nano-electrospray ion source (Thermo Fisher Scientific). The MS methods included a full MS scan at a resolution of 70,000 followed by 10 data-dependent MS2 scans at a resolution of 17,500. An MS scan range of 350 to 1400 m/z was selected and precursor ions with unassigned charge states, a charge state +1, or a charge state of greater than +8 were excluded. The peak list-generating software used in this study was Proteome Discoverer™ Software 2.1 (Thermo Fisher Scientific). The SEQUEST-HT search engine was employed in the data processing. The MS raw data files were searched against *P. aeruginosa* (strain ATCC 15692/DSM 22644/CIP 104116/JCM 14847/LMG 12228/1C/PRS 101/PAO1) Database (1435 proteins, March 2022) and a list of common protein contaminants (www.thegpm.org/crap/) (accessed on 27 May 2022). The Proteome Discoverer decoy database together with the Percolator algorithm was used to calculate the false-positive discovery rate of the identified peptides based on Q-values which were set to 1%. The Precursor Ions Quantifier node in Proteome Discoverer™ Software was employed to quantify the relative MS signal intensities of dimethyl labeled peptides. The control channels were used as denominators to generate abundance ratios of C_PACL/PACL. Log2 of the normalized ratio was used to calculate the mean and standard deviation of fold change across all three biological replicates. The proteomic data were submitted to the ProteomeXchange consortium (PXD034530).

### 4.9. Fibroblast and Macrophage Experiments

Responsiveness of human fibroblasts and macrophages against *P. aeruginosa* parent strain (PACL) and Chlorhexidine (CHG)-treated strain (C_PACL) planktonic and sessile forms was evaluated. Primary human dermal fibroblasts (isolated from healthy male neonatal foreskin tissues; Protocol number IRB 120/63 approved by the Institutional Review Board of the Faculty of Medicine, Chulalongkorn University) were grown in Dulbecco’s Modified Eagle Medium (DMEM) (Gibco, Life Technologies, Ltd., Paisley, Scotland, UK) supplemented with 10% fetal bovine serum (FBS) (Gibco, Life Technologies, Ltd.) in a humidified atmosphere at 37°C, 5% CO_2_. The primary human dermal fibroblasts (1 × 10^5^ cells/well) in 12-well culture plates were incubated with either the *Pseudomonas* planktonic bacteria (1 × 10^4^ CFU) or the sessile form, prepared by 1 × 10^4^ CFU of bacteria on the polystyrene plates, at 37°C in the CO_2_ incubator for 6 h. The protocol of primary fibroblasts isolation from a publication was followed [62]. Briefly, the foreskin tissues from healthy volunteers (IRB 120/63) were collected in sterile containers supplemented with DMEM and 10% FBS with 5 µg/mL of Plasmocin (the mycoplasma elimination reagent) (InvivoGen, San Diego, CA, USA). Then, the tissues were minced and incubated on DMEM with 20% PBS plus gentamicin (Sigma-Aldrich) at 37 °C in the CO_2_ incubator. The skin remnants were discarded after 7 days of incubation with the presence of dermal fibroblasts (observed by light microscopy analysis). At 21 days of incubation, the dermal fibroblasts were harvested with 0.25% trypsin in EDTA (Ethylenediaminetetraacetic acid) before centrifugation. The pellets of fibroblasts were resuspended in DMEM with 10% PBS plus gentamicin (Sigma-Aldrich) for quantification using a hemocytometer. In parallel, THP-1 (ATCC TIB202), a human monocytic cell line, was cultured in RPMI 1640 (Gibco Life Technologies, Ltd.) supplemented with 10% FBS before the differentiation into macrophages using 100 nM phorbol myristate acetate (Sigma-Aldrich). The THP-1-derived macrophages, seeded at 1 × 10^5^ cells/well in 12-well plates, were treated with planktonic or sessile forms of *P. aeruginosa* as described in the fibroblast experiments. After treatment, the supernatant was collected and determined for cytokine concentration using ELISA assays. The cytokine and Toll-like receptor gene expressions were determined by qRT-PCR using primers listed in Appendix A with the previously mentioned protocol (in the section on bacterial gene expression). Briefly, total RNA was extracted and converted to cDNA before determination of the expression of genes for cytokines and Toll-like receptors (normalized with β-actin) using the 2^−ΔΔct^ method.

### 4.10. Statistical Analysis

All data were analyzed using the Statistical Package for Social Sciences software (SPSS 22.0, SPSS Inc., Chicago, IL, USA) and Graph Pad Prism version 7.0 software (La Jolla, CA, USA). Results were presented as mean ± standard error (SE). The differences between two groups and multiple groups were examined for statistical significance by an unpaired *t*-test or one-way analysis of variance (ANOVA) with Tukey’s analysis, respectively. The survival analysis and time-point data were determined by the Log-rank test and repeated measures ANOVA, respectively. A *p*-value < 0.05 was considered statistically significant.

## 5. Conclusions

With prolonged exposure to sub-lethal Chlorhexidine, *P. aeruginosa* developed a cross-resistance to several antibiotics, especially colistin, possibly through the release of Ca^2+^ from bacterial membrane due to Chlorhexidine replacement, which initially triggered LadS/GacSA (c-di-GMP-independent) followed by SiaD-induced-c-di-GMP-dependent *Psl*-mediated aggregation causing small-colony variants, metabolic arrests, and reduced inflammatory activation (on fibroblasts and macrophages). However, the Chlorhexidine-mediated *Psl*-predominant *P. aeruginosa* demonstrated a similar wound infection to the parent strain with the cross-resistance to some antibiotics with a possible significance. Hence, we propose adequate doses and frequency of Chlorhexidine use.

## Figures and Tables

**Figure 1 ijms-23-08308-f001:**
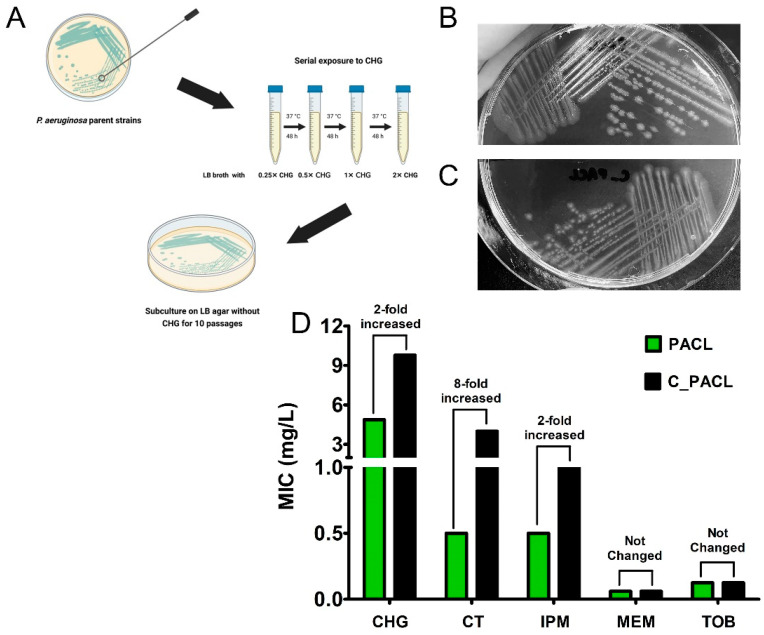
Diagram of Chlorhexidine (CHG) treatment protocol in *P. aeruginosa* clinical isolates demonstrates the subsequent increase in CHG concentrations during the treatment (**A**). The representative pictures of the colonies in CHG-free Luria-Bertani (LB) agar for 24 h at 37 °C of *P. aeruginosa* parent strain (PACL) (**B**) and CHG-treated strain (C_PACL) (**C**) indicates small colony variants (SCVs) in C_PACL. Minimal inhibitory concentrations (MICs) using broth microdilution method of PACL and C_PACL toward CHG and other antibiotics (CT: colistin, IPM: imipenem, MEM: meropenem, and TOB: tobramycin) are demonstrated (**D**). Triplicated independent experiments were performed.

**Figure 2 ijms-23-08308-f002:**
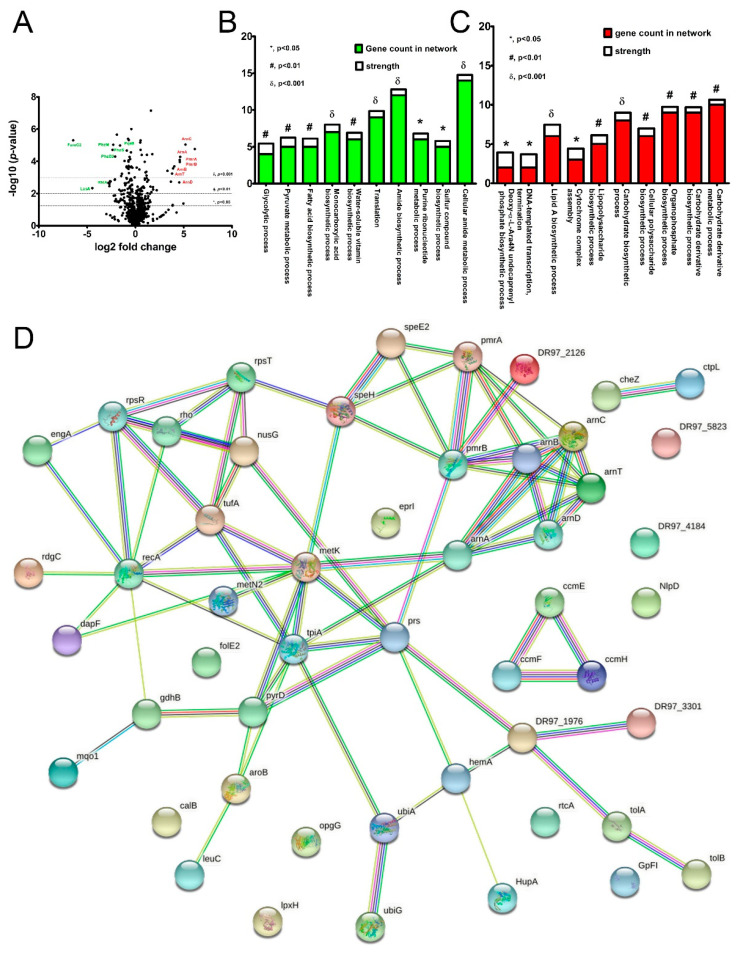
Proteomic study of the biofilms from *P. aeruginosa* parent strain (PACL) and Chlorhexidine (CHG)-treated strain (C_PACL). The volcano plot displayed the distribution of 957 proteins in which the protein abundance (log_2_ fold change of C_PACL/PACL) plotted with the significant values (-log_10_ *p*-value) (**A**). Proteins in the biological process that downregulated (**B**) and upregulated (**C**) in C_PACL compared to PACL (*p* < 0.01) using STRING v.10 are demonstrated. The functional protein association networks (**D**) for the complicated connection of upregulated proteins (*p* < 0.01) by STRING v.10 are also showed.

**Figure 3 ijms-23-08308-f003:**
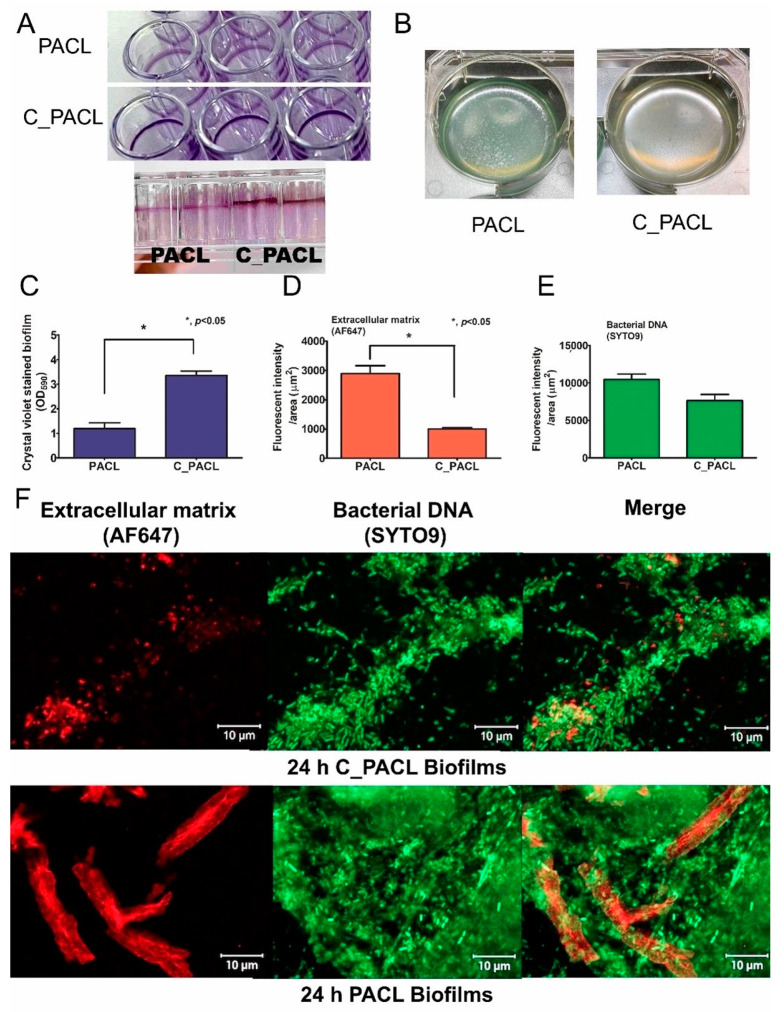
Biofilm characteristic of *P. aeruginosa* parent strain (PACL) and Chlorhexidine (CHG)-treated strain (C_PACL). The representative pictures of biofilms at the solid–liquid interface in 96-well polystyrene plates stained with crystal violet (the top and side views) (**A**) and the pellicle biofilms after incubation at the static condition for 24 h at 37 °C (biofilms at the liquid-air interface) (**B**) are demonstrated in 6-well polystyrene plates. The intensity of crystal violet stain from 24 h biofilm in 96-well plates (**C**), and intensity of fluorescent stains from 24 h biofilm on cover glasses of *P. aeruginosa* for extracellular matrix (ECM) by AF647 (red color fluorescence) (**D**) or bacterial nucleic acid by SYTO9 (green color fluorescence) (**E**), and the representative fluorescent-stained pictures (**F**) are demonstrated. Mean ± SEM are presented with the unpaired *t*-test analysis (*, *p* ˂ 0.05). The representative fluorescent images of biofilms on glass coverslips stained for ECM by AF647 and bacterial nucleic acid by SYTO9 (**F**) are also demonstrated.

**Figure 4 ijms-23-08308-f004:**
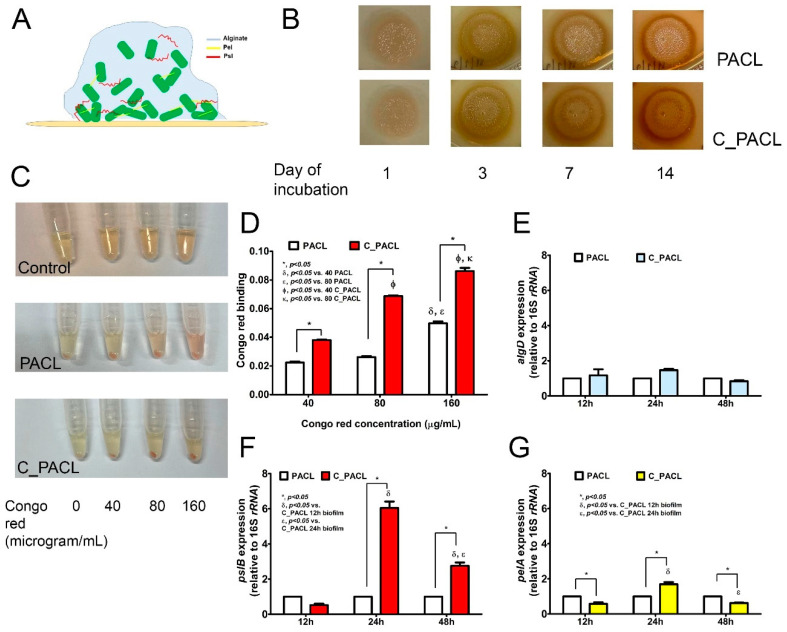
Exopolysaccharide (EPS) produced by *P. aeruginosa* parent strain (PACL) and Chlorhexidine (CHG)-treated strain (C_PACL). The graphic indicates alginate as a major component of EPS in *P. aeruginosa* mucoid biofilms, whereas Pel and *Psl* maintains biofilm strength (**A**). Colony with biofilms on Luria Broth (LB) agar supplemented with Congo red color (a color for staining of Pel and *Psl* in EPS) (**B**), Congo red binding activity of planktonic cells in LB broth supplemented with Congo red (**C**), the Congo red binding activity quantified by the remaining Congo red in the broth (see method) (**D**) and the expressions of *algD* (**E**), *pslB* (**F**), and *pelA* (**G**), the encode proteins for alginate, *Pel*, and *Psl* synthesis, as determined by qRT-PCR, are demonstrated. The experiments were performed in independent triplicate. Mean ± SEM are presented with the one-way ANOVA followed by Tukey’s analysis (*, *p* ˂ 0.05; ^Φ^, *p* ˂ 0.05; ^δ^, *p* ˂ 0.05; ^ε^, *p* < 0.05; and ^κ^, *p* < 0.05).

**Figure 5 ijms-23-08308-f005:**
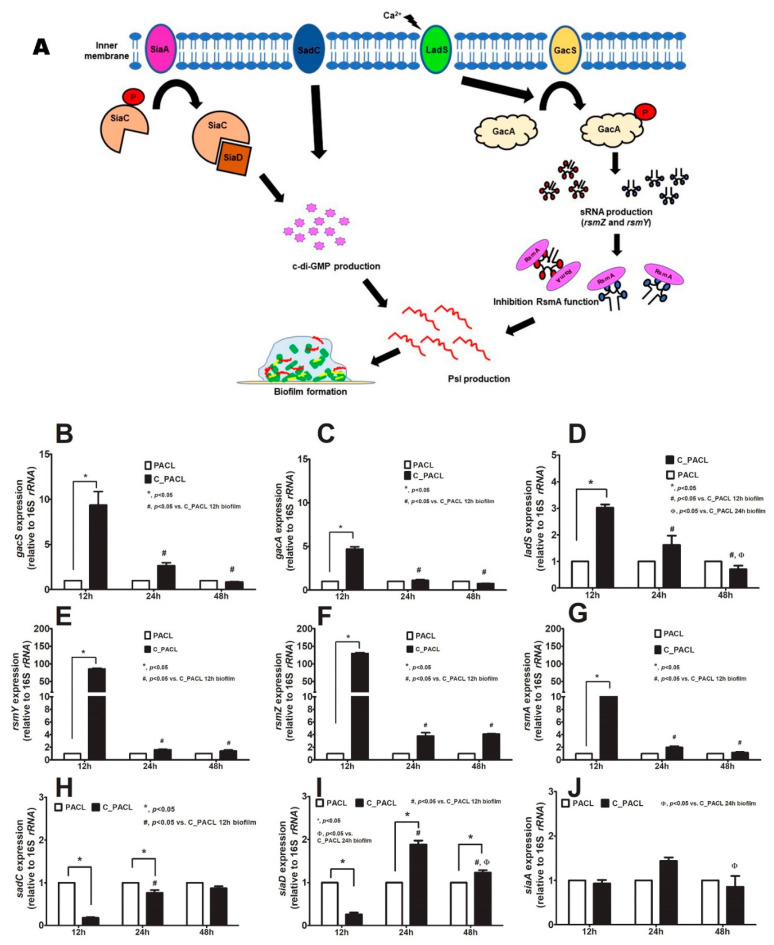
Diagram of *Psl* biosynthesis in *P. aeruginosa* with the bis-(3′-5′)-cyclic dimeric guanosine monophosphate (c-di-GMP)-dependent, using several diguanylate cyclases (SadC, SiaD, and SiaAC system) (left side) and c-di-GMP-independent pathways (using RsmA transcriptional regulator through *rsmZ*, *rsmY*, and LadS/GacSA) is demonstrated (**A**). Gene expression profiles in *P. aeruginosa* parent strain (PACL) and Chlorhexidine (CHG)-treated strain (C_PACL) biofilms in the c-di-GMP-independent pathway, including *gacS* (**B**), *gacA* (**C**), *ladS* (**D**), *rsmY* (**E**), *rsmZ* (**F**), and *rsmA* (**G**), and the c-di-GMP-dependent pathway, including *sadC* (**H**), *siaD* (**I**), and *siaA* (**J**) as determined by qRT-PCR are demonstrated (the experiments were performed in independent triplicate). Mean ± SEM are presented with the one-way ANOVA followed by Tukey’s analysis (*, *p* ˂0.05; ^#^, *p* ˂ 0.05; and ^Φ^, *p* ˂ 0.05).

**Figure 6 ijms-23-08308-f006:**
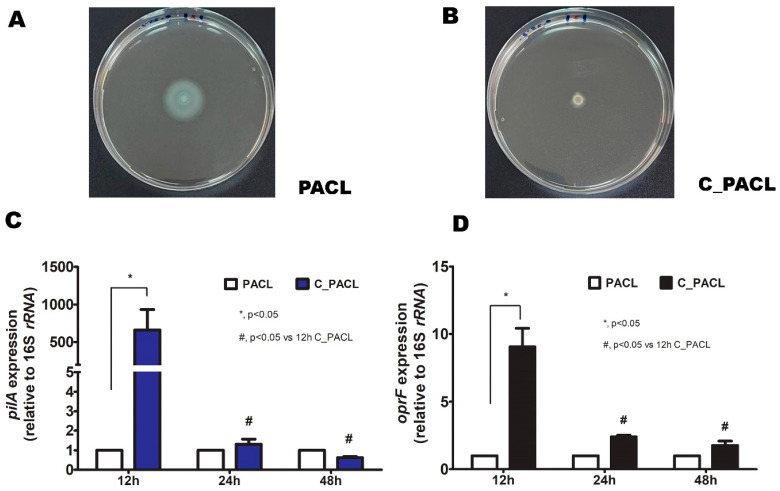
The characteristics of *P. aeruginosa* parent strain (PACL) and Chlorhexidine (CHG)-treated strain (C_PACL) biofilm as indicated by the reduction of motility (swarming motility) using 0.5% agar in Luria Broth (LB) agar medium by spotting the overnight culture of bacteria at the center of the plates after 18 h incubation at 37 °C (**A**,**B**) with the expressions of *pilA* (**C**) and *oprF* (**D**) (encoding for type-4 pili biosynthesis and OprF proteins of bacterial appendages, respectively) are demonstrated (the experiments were performed in independent triplicate). Mean values of the relative mRNA expression were plotted with error bars representing the SEM. The *p*-values were calculated using one-way ANOVA followed by Tukey’s analysis (*, *p* ˂ 0.05 and ^#^, *p* < 0.05 considered statistically significant).

**Figure 7 ijms-23-08308-f007:**
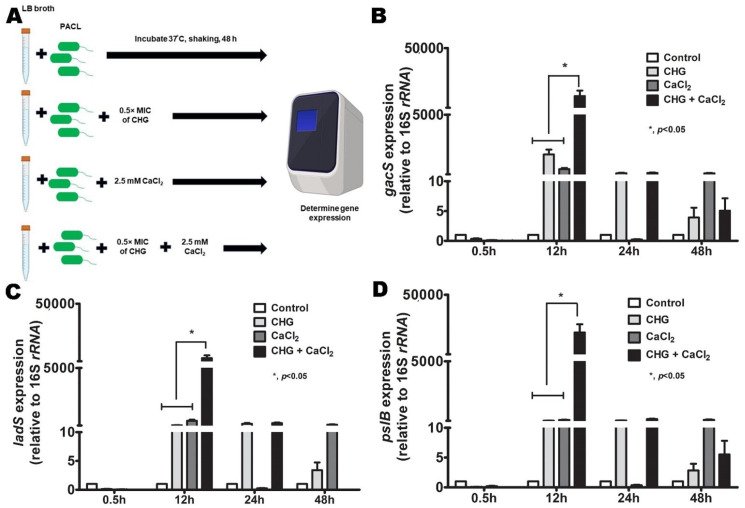
LadS/GacS regulation system in the presence of Ca^2+^. The diagram of the experiments that *P. aeruginosa* parent strain (PACL) treated with Chlorhexidine (CHG) alone or in the combination with CaCl_2_ before an evaluation of gene expression by qRT-PCR (**A**) of *gacS* (**B**), *ladS* (**C**), and *pslB* (**D**) are demonstrated (the experiments were performed in independent triplicate). Mean ± SEM are presented with the one-way ANOVA followed by Tukey’s analysis (*, *p* ˂ 0.05 considered statistically significant).

**Figure 8 ijms-23-08308-f008:**
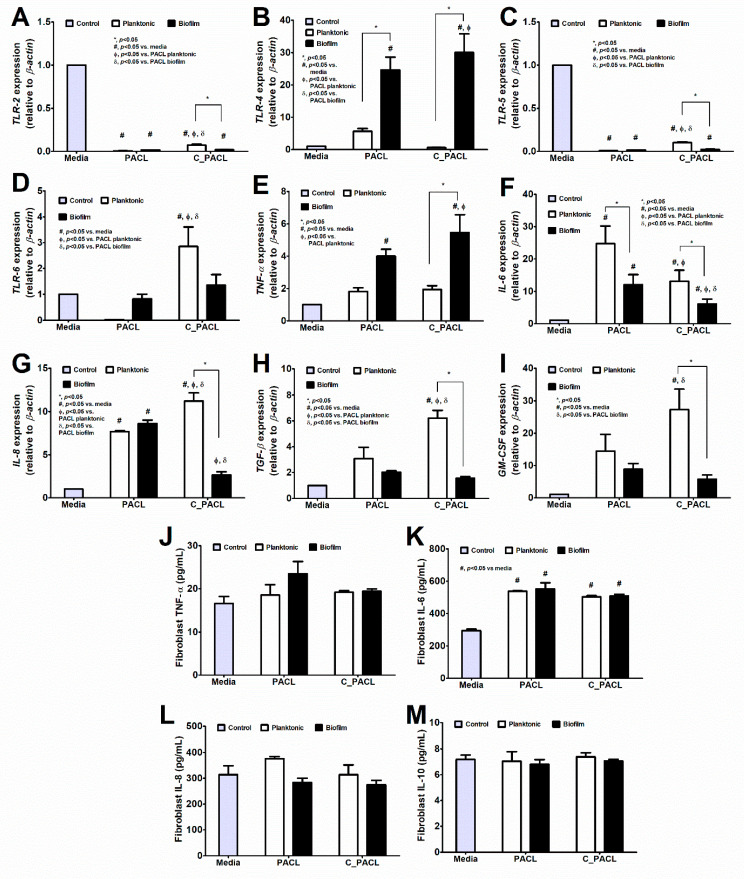
The responses of fibroblast toward *P. aeruginosa* parent strain (PACL) and Chlorhexidine (CHG)-treated strain (C_PACL) in planktonic (non-biofilms) and sessile (biofilms) forms. The gene expressions by qRT-PCR of *TLR-2*, *TLR-4*, *TLR-5,* and *TLR-6* (**A**–**D**) and pro-inflammatory cytokines (*TNF-α*, *IL-6*, *IL-8*, *TGF-β*, and *GM-CSF* (**E**–**I**) with supernatant cytokines using ELISA (*TNF-α*, *IL-6*, *IL-8*, and *IL-10*) (**J**–**M**) are demonstrated. The experiments were performed in independent triplicate. Mean ± SEM are presented with the one-way ANOVA followed by Tukey’s analysis (*, *p*˂0.05; ^#^, *p* < 0.05; ^Φ^, *p* < 0.05; and ^δ^, *p* < 0.05 considered statistically significant).

**Figure 9 ijms-23-08308-f009:**
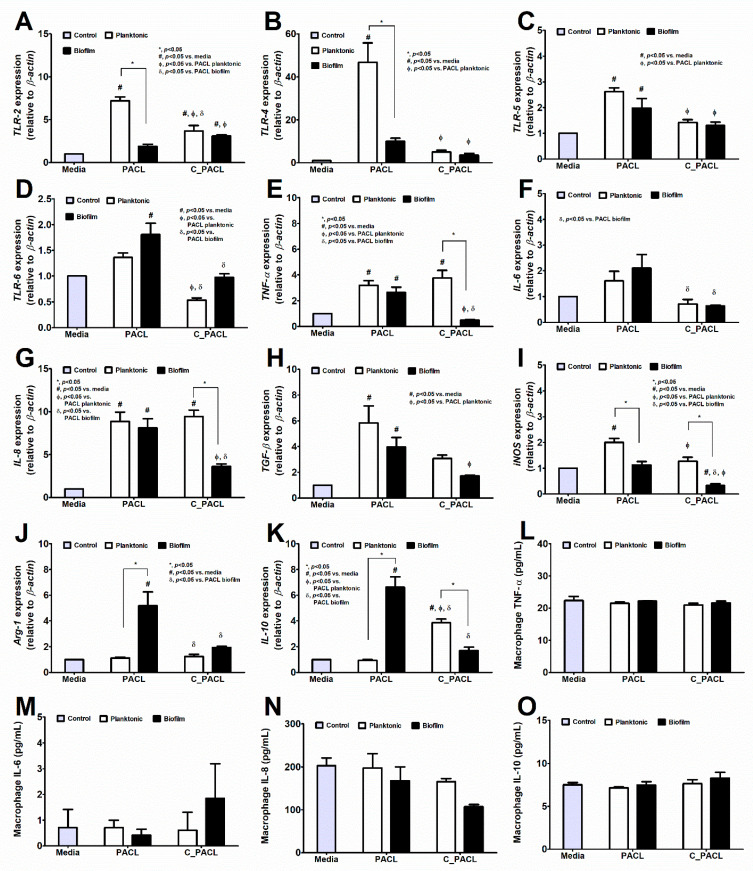
The responses of THP-1-derived macrophage toward *P. aeruginosa* parent strain (PACL) and Chlorhexidine (CHG)-treated strain (C_PACL) in planktonic (non-biofilms) and sessile (biofilms) forms. The gene expressions by qRT-PCR of *TLR-2*, *TLR-4*, *TLR-5,* and *TLR-6* (**A**–**D**), and cytokines (*TNF-α*, *IL-6*, *IL-8*, and *TGF-β*) (**E**–**H**), pro-inflammatory (*iNOS*) (**I**), and anti-inflammatory molecules (*Arg-1* and *IL-10*) (**J**,**K**) with supernatant cytokines using ELISA (TNF-α, IL-6, IL-8, and IL-10) (**L**–**O**) are demonstrated. The experiments were performed in independent triplicate. Mean ± SEM are presented with the one-way ANOVA followed by Tukey’s analysis (*, *p*˂0.05; ^#^, *p* < 0.05; ^Φ^, *p* < 0.05; and ^δ^, *p* < 0.05 considered statistically significant).

**Figure 10 ijms-23-08308-f010:**
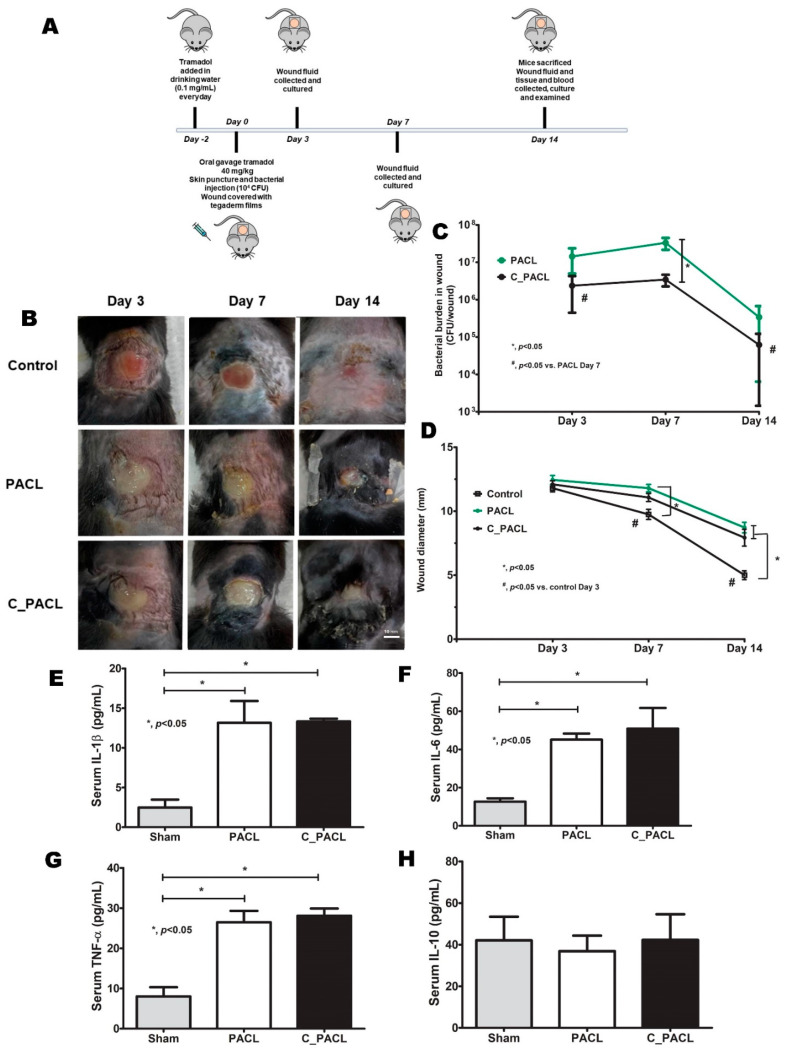
The characteristics of wounds from *P. aeruginosa* parent strain (PACL) and Chlorhexidine (CHG)-treated strain (C_PACL); control (n = 8), PACL (n = 9), and C_PACL infection (n = 8), as indicated by diagram of experiments (**A**), the representative pictures of wounds (**B**), bacterial burdens in wounds (**C**), wound diameters (**D**), and the serum cytokines (*IL-1β*, *IL-6*, *TNF-α*, and *IL-10*) at 14 days of experiment (**E**–**H**) are demonstrated. Mean ± SEM are presented with the one-way ANOVA followed by Tukey’s analysis (*, *p* ˂ 0.05 and ^#^, *p* < 0.05 considered statistically significant).

**Figure 11 ijms-23-08308-f011:**
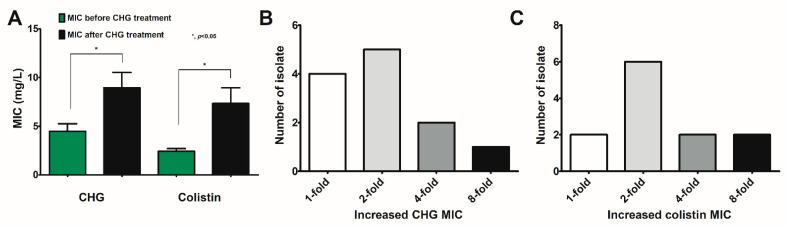
Chlorhexidine (CHG) treatment in other 12 clinical isolates of *P. aeruginosa*. Minimal inhibitory concentrations (MICs) of CHG-treated strains toward CHG and colistin using broth microdilution method (**A**) and the distribution of strains with increased MICs against CHG and colistin are demonstrated (**B**,**C**). Mean ± SEM are presented with the unpaired *t*-test (*, *p* ˂ 0.05 considered statistically significant).

## Data Availability

The data presented in this study are available on request from the corresponding author.

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
