# Peer review of "Chlorhexidine Promotes Psl Expression in Pseudomonas aeruginosa That Enhances Cell Aggregation with Preserved Pathogenicity Demonstrates an Adaptation against Antiseptic"

_ijms, 2022, doi:10.3390/ijms23158308_

Round 1
Reviewer 1 Report
The present paper approaches the adaptation of Pseudomonas aeruginosa following in vitro contact with Chlorexidine. The Chlorexidine adapted Ps. aeruginosa showed different colonies, metabolic downregulation and resistance against colistin partly through the modification of the LPS. Furthermore, the adapted strain formed different biofilm partly due to the overexpression of psl , impacted also by the addition of Ca++, through LadS/GacSa pathway maintaining the aggregation with SiaD-mediated c-di-GMP dependance . The addition of the modified strain induced on fibroblast and macrophages a lower expression of Il-6 and Il-8 whereas the wound diameters and bacterial burden of infected mouses were similar to those produced by the original strain. Finally, Authors recommended adequate doses and frequency of chlorexidine use.The paper appears full of information and well write providing interesting informations; the clarification of some details may lead to its publication.
Line 482-485: the statement is not clear, rewrite
Line 489: the mouse serum cytokine evaluation (line 359) is not described
Line 505, 512, 528, 540, 551, 557, 587: it is not mentioned which/how strains of Ps. aeruginosa were used
Line 602 provide more information about the qRT-PCR analysis
Reviewer 2 Report
Singkham-In et al studied the ability of P. aeruginosa to acquire resistance toward chlorhexidine.
Some aspects need to be clarified.
Figure 3 should be improved. By figure 3A, the biofilm seems to be just around the well.
In figure 3F, the resolution should be improved, as well as the focus.
In vivo studies description, the ethical committee approval is missing and the housing conditions and information regarding mice (supplier, SPF or not).
Histological analysis should be performed in order to demonstrate the effect on tissue.
The authors should clarify how they ensured that bacterial infection in the wound is only due to P. aeruginosa infection, or an infection site can have a co-infection with other bacteria, e.g Staphylococcus spp?
Do you use a selective or differential medium to isolate and identify only P.aeruginosa in wound infection?
The authors should be improved the collection method and analysis of in vivo fluid material.
In Material and methods, in confocal description is missing the magnification and confocal parameters used.
In the 4.9 section, the authors did not specify the fibroblasts cell line used.
Reviewer 3 Report
The manuscript describes the use of chlorhexidine as a model antiseptic in adaptation to a mammalian host. The animal model used is murine which is not fully adequate for translating into the human host.
In the work, the authors examine the low dosage use of chlorhexidine on the type of bacterial growth, with or without biofilm. The authors observed increased colistin resistance in bacteria treated with chlorhexidine as opposed to the control, which was attributed to the mechanism of action requiring a positive charge of membranes in both cases. The more detailed question as to the mechanism was also examined by looking into gene expression.
Overall, the manuscript presents new data on the adaptation of P. aeruginosa to a mammalian host although some questions remain unanswered.
There are some places where professional editing would be beneficial to the manuscript.
Round 2
Reviewer 2 Report
I accept the manuscript in its present form. The authors responded to all questions and the corrections were included in the manuscript.